# Attentional enhancement predicts individual differences in visual working memory under go/no-go search conditions

**Daniel Tay** *, **John J. McDonald**

Department of Psychology, Simon Fraser University, Burnaby, British Columbia, Canada

* daniel_tay@sfu.ca

**Data Availability Statement:** All quantitative observations summarized in Figs 2¬–4 and S1 are available at https://osf.io/4wdzq.

## Abstract

Attention-control processes transfer relevant information to visual working memory (WM) and prevent irrelevant information from consuming WM resources. Although event-related potentials (ERPs) have revealed attention-control processes associated with enhancement of relevant stimuli (targets) and suppression of irrelevant stimuli (distractors), only the suppressive processes have been found to predict WM capacity. We hypothesised a link between target-enhancement processes and WM capacity would be revealed in a task that requires more control than the conventional visual search paradigms used to study target selection. Here, participants searched for a pop-out target on Go trials and withheld responses on an equal number of randomly intermixed No-Go trials, depending on the colour of the stimulus array. Magnitudes of ERP indices associated with target enhancement (the singleton detection positivity, SDP, and N2pc) were positively correlated with individual differences in WM capacity. These relationships vanished when participants searched for the pop-out target on every trial, regardless of stimulus-array colour. Inhibitory processes associated with suppressing distractors ($P_D$) and withholding responses (no-go P3) on No-Go trials did not predict WM capacity. These findings indicate that target-enhancement mechanisms control access to WM in search tasks that require dynamic control and disconfirm the view that the gateway to WM is entirely inhibitory by nature.

## Introduction

Neurologically healthy young adults can remember up to 3 or 4 visual objects for short periods of time (1 to 3 seconds) without rehearsal [1–3]. The precise capacity limit of this type of short-term working memory (WM) varies across individuals, and these individual differences are predictive of performance on tasks that measure higher-order cognitive abilities and fluid intelligence [4–8]. The associations between WM capacity and higher-order cognitive abilities are more apparent in the face of task-irrelevant sources of information that have the potential to distract individuals from the task at hand. This observation has led to the view that individual differences in attentional capabilities contribute substantially to differences in WM capacity [9,10]. Consistent with this general controlled-attention view of WM capacity, many

**Funding:** This study was supported by the Natural Sciences and Engineering Research Council of Canada (to JJM, RGPIN-2015-05095) and the Canada Research Chairs program (to JJM, 950-230768). The funders had no role in study design, data collection and analysis, decision to publish, or preparation of the manuscript.

**Competing interests:** The authors have declared that no competing interests exist.

**Abbreviations:** BF, Bayes factor; CDA, contralateral delay activity; EEG, electroencephalogram; ERP, event-related potential; HEOG, horizontal electrooculogram; N2pc, posterior contralateral N2; P2a, anterior P2; $P_D$, distractor positivity; RT, response time; SDP, singleton detection positivity; WM, working memory.

researchers believe that access to WM is governed by inhibitory attention processes that actively filter out irrelevant distractors [11–14].

Event-related potentials (ERPs) and other non-invasive neuroscientific methods have been used to investigate the neural processes involved in WM as well as the attention processes controlling access to WM. Such methods have been used to isolate visual WM activity that occurs between presentations of an initial array of to-be-remembered items and a subsequent test array. Participants in these change-detection tasks are instructed to indicate whether the test array is identical to the memory array or whether one item differs between the 2 arrays. ERP waveforms that are time-locked to the initial memory array reveal lateralized activity over the posterior scalp when participants are instructed to detect changes on one side of the array or the other (specified at the start of each trial with a symbolic cue). The magnitude of this contralateral delay activity (CDA) initially increases when the number of to-be-remembered items (set size) is increased but reaches asymptote when the set size is equal to, or greater than, the individual's visual WM capacity (estimated in a different task) [15]. Thus, the CDA appears to reflect activity associated with items being maintained in WM. Interestingly, when the cued visual hemifield contains 2 relevant items and 2 irrelevant (i.e., to-be-ignored) items, the CDA is actually larger for low-capacity individuals than it is for high-capacity individuals [16]. This counter-intuitive pattern of results suggests that high-capacity individuals manage to filter out the irrelevant items, thereby preventing their active maintenance in WM, whereas low-capacity individuals fail to do so. The findings are also consistent with the view that individual differences in WM capacity reflect how efficiently an individual can prevent irrelevant information from inadvertently reaching WM systems.

ERP data obtained from visual search and change-detection tasks have provided converging evidence for the filtering-efficiency hypothesis of WM capacity [17,18]. In both tasks, targets elicit an ERP component called the posterior contralateral N2 (N2pc), which has been hypothesised to reflect an early stage of attention selection [19], while salient distractors elicit an ERP component called the distractor positivity ($P_D$), which has been hypothesised to reflect suppression of irrelevant and potentially distracting visual objects [20]. These components typically begin 200 to 250 ms after stimulus onset and approximately 100 ms before the CDA begins. This temporal sequence suggests that the N2pc and $P_D$ reflect target- and distractor-centered selection processes, respectively, that occur before WM maintenance. Critically, however, only the $P_D$ magnitudes have been found to correlate with WM capacities across individuals, with larger magnitudes being predictive of higher capacities. No such association has been found between the target-elicited N2pc and WM capacity [17,18,21]. Together, these findings indicate that individual differences in WM capacity depend primarily on the ability to suppress irrelevant visual information, not on the ability to selectively enhance relevant information.

The main purpose of the present study was to further test for a link between attentional enhancement of target processing and individual differences in visual WM capacity. This is important because conclusions about the lack of such a link are based on a small number of null results that might not generalise to other experimental conditions. Moreover, from a theoretical perspective, attention-control processes that enhance relevant information could contribute to the ability to maintain focus on current goals and other sources of relevant information in WM [22–24]. Here, we hypothesised that target selection may have been too automatic in prior visual search studies to reveal such a link. This hypothesis was premised on the distinction between automatic and controlled processing [25] and on previous studies indicating that performance differences between low- and high-capacity individuals emerge only in tasks that require controlled processing [23,26,27]. In terms of the basic processing distinction, researchers theorised that higher-level cognitive commands that are required to

initiate an attention operation initially require considerable control but become routine with sufficient practice so that they can be executed automatically [25,28]. Consistent with this theoretical perspective, the cognitive command to selectively enhance a task-relevant stimulus may become automated across a wide range of visual-search tasks, including ones in which the target does not "pop out" from the rest of the array [29,30]. Thus, we surmised that target-enhancement processes will be predictive of the individual differences in visual WM only when such automation is prevented.

In Experiment 1, we introduced a Go/No-Go aspect to an otherwise typical pop-out search task to disrupt the automation of target selection. Healthy young adults (*n* = 44) viewed displays containing 16 blue lines or 16 yellow lines (**Fig 1**). On half the trials, the lines were all horizontal or all vertical. On the remaining trials, one of the lines was rotated 90 degrees from the rest. Participants were instructed to indicate the presence or absence of a uniquely oriented line (i.e., the singleton) on relevant-colour trials and to refrain from responding on irrelevant-colour trials (herein called Go trials and No-Go trials, respectively). The orientations of the singleton and the surrounding items swapped randomly from trial to trial to discourage the involvement of suppressive attention mechanisms that filter out nontargets [19]. Thus, search was presumed to be accomplished by selectively enhancing the target. Based on the results of a recent study using this design [31], we expected attentional enhancement of the singleton to occur on Go trials but to be prevented on No-Go trials.

Two ERP components were used to track target-enhancement processes. First, a positivity with bilateral maxima over the occipital scalp was isolated by subtracting singleton-absent ERPs from singleton-present ERPs. This singleton detection positivity (SDP) begins approximately 200 ms after display onset and appears to be associated with the detection of task-relevant singletons [31,32]. Second, a contralateral negativity called the N2pc was isolated over the posterior scalp by subtracting ERPs recorded ipsilaterally with respect to the target singleton's location from the corresponding contralateral ERPs. The N2pc typically occurs 170 to 350 ms after display onset and, as noted previously, is associated with the focusing of attention on individual search items [19]. The N2pc is evident when target and nontarget features swap randomly to prevent suppressive filtering, thereby linking the N2pc to target enhancement rather than distractor suppression [31]. The singleton was expected to elicit the SDP and N2pc on Go trials and little to no such activities on No-Go trials [32].

We measured 3 additional ERP components that were expected to occur in Experiment 1 to determine whether other processes in this modified visual-search task were predictive of individual differences in visual WM. One of these components, the anterior P2 (P2a) [33], was isolated over the prefrontal scalp by subtracting ERPs elicited by No-Go trials from ERPs elicited by Go trials. The P2a typically occurs 180 to 300 ms after display onset and has been associated with detection of relevant stimuli. In Experiment 1, Go displays were expected to elicit the P2a, whether or not they contained a singleton [32]. Another one of these components, the $P_D$ [20], was isolated over the posterior scalp by subtracting ERPs ipsilateral to the distractor's (i.e., singleton on No-Go trials) location from the corresponding contralateral ERPs. The $P_D$ typically occurs 200 to 500 ms after display onset and is associated with suppression of sensory inputs from distractor locations [34,35]. However, the $P_D$ elicited on No-Go trials occurs relatively late, suggesting that suppression mechanisms on No-Go trials prevent access to WM and not the orienting of attention in this paradigm [32]. Finally, a positivity called the no-go P3 [36] was isolated over the central scalp by subtracting ERPs elicited on No-Go trials from ERPs elicited on Go trials. The no-go P3 typically occurs 200 to 500 ms after display onset and has been associated with inhibition of manual responses on No-Go trials [37].

singleton
absent

singleton
present

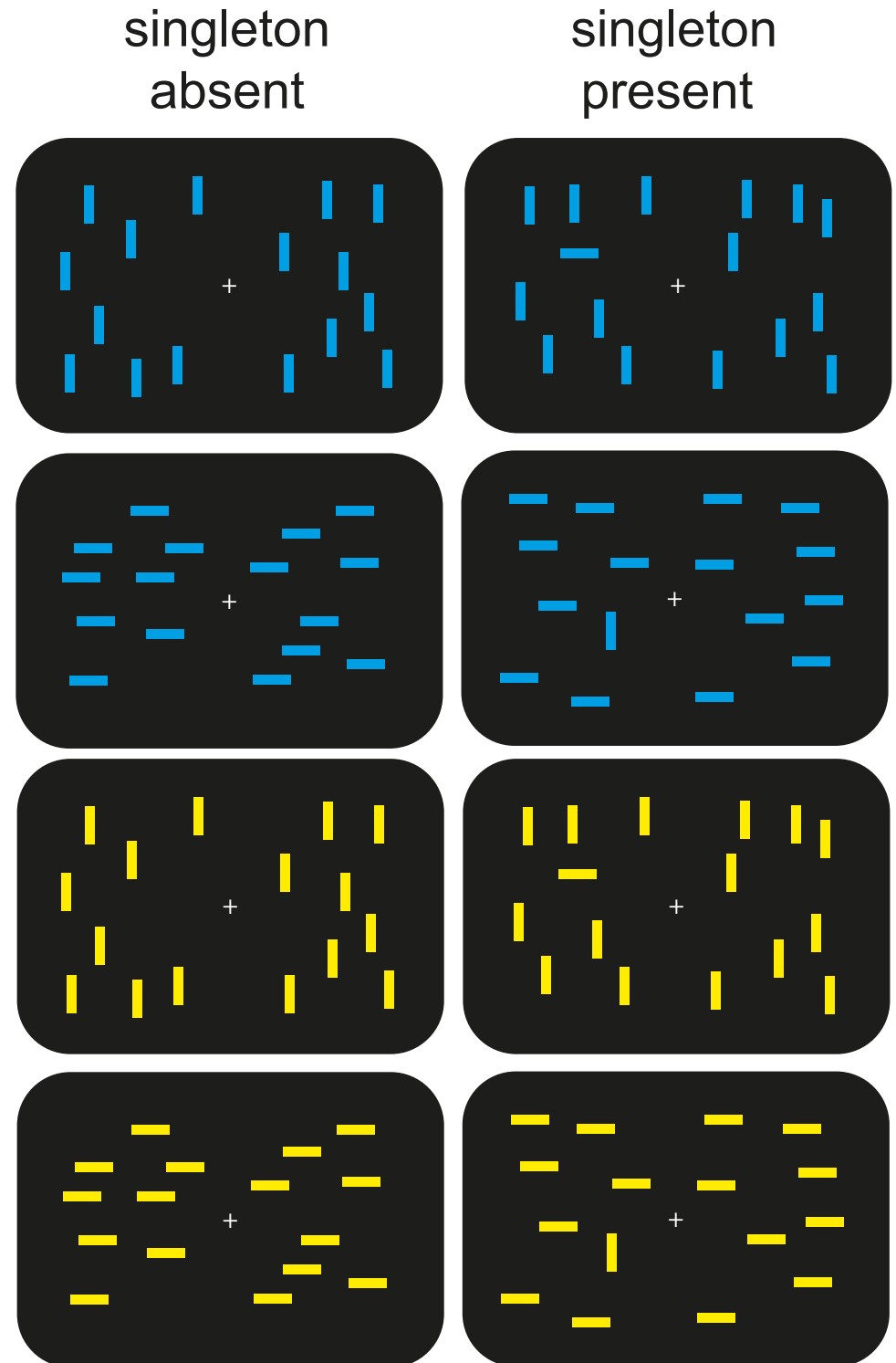

**Fig 1. Example stimulus displays used in Experiments 1 and 2.**

## Results

### ERPs reveal time course of stimulus processing on Go and No-Go trials

The ERP activities related to early detection of task relevance (P2a), subsequent selective target enhancement (N2pc and SDP), and late distractor suppression (P_D and no-go P3) unfolded in the expected sequence. Starting approximately 150 milliseconds after the appearance of a search array, ERP waveforms recorded over the frontal scalp became more positive on Go trials than on No-Go (P2a) (**Fig 2A**). This difference in mean amplitude (4.32 µV) was found to be statistically significant in the P2a measurement window (186 to 236 ms), $t(43) = 17.12$, $p < 0.001$, $d = 2.58$. Approximately 50 ms later, ERP waveforms recorded at lateral occipital electrodes became more positive on target-present Go trials than on target-absent Go trials (SDP) (**Fig 2B and 2C**). This mean-amplitude difference (3.39 µV) was statistically significant

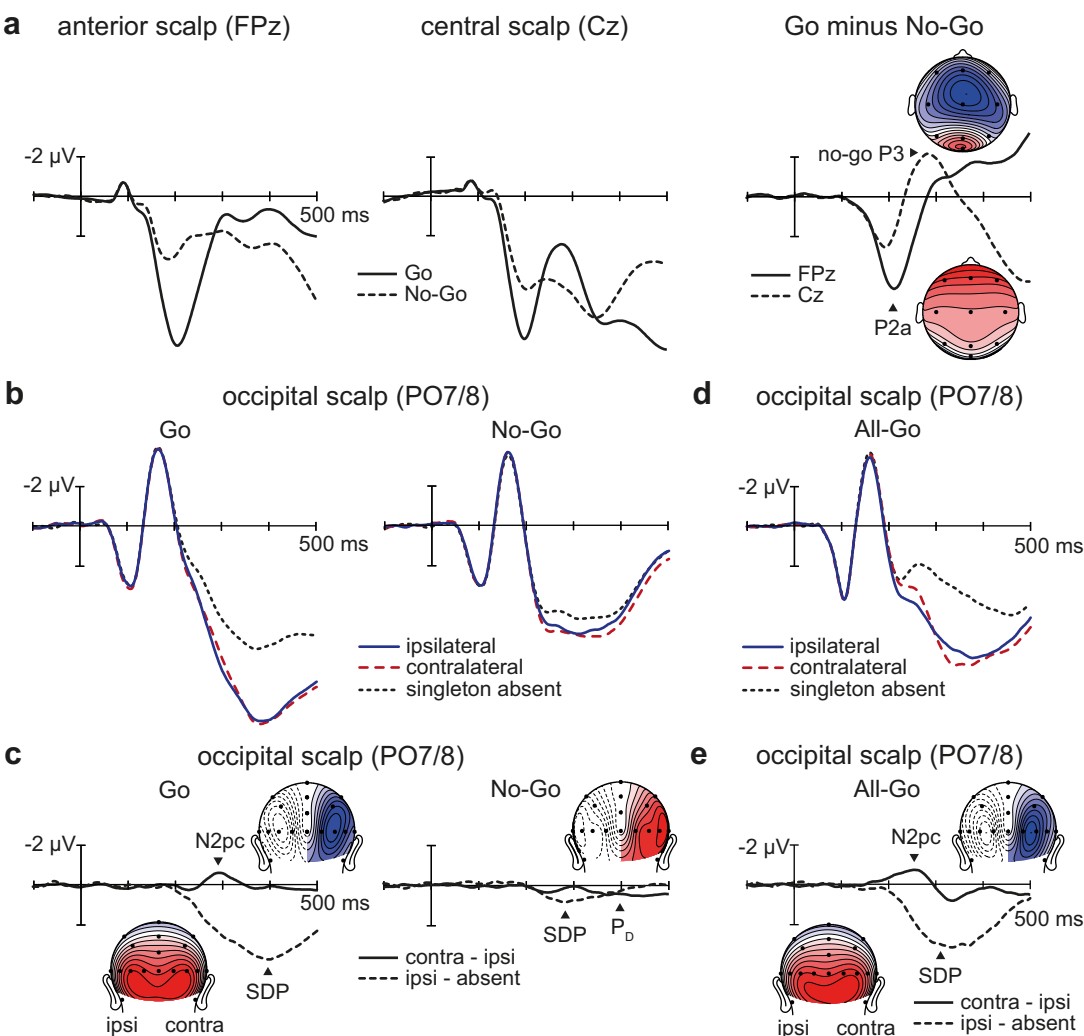

**Fig 2. Grand-averaged ERP results from Experiments 1 and 2.** Negative voltages are plotted upwards by convention. The underlying data supporting this figure can be found at https://osf.io/4wdzq. (**a**) Go and No-Go ERPs and associated Go-minus-No-Go difference waveforms from Experiment 1, plotted at frontal (FPz) and central (Cz) scalp sites. (**b**) Occipital ERPs plotted separately for Go and No-Go trials of Experiment 1. (**c**) Difference waveforms over the occipital scalp from Experiment 1. (**d**) All-Go ERPs over the occipital scalp from Experiment 2. (**e**) Difference waveforms over the occipital scalp from Experiment 2. ERP, event-related potential.

in the SDP measurement window (318 to 418 ms), $t(43) = 14.97$, $p = < 0.001$, $d = 2.26$. On target-present trials, the contralateral occipital waveform was more negative than the ipsilateral occipital waveform in the time range of the N2pc (274 to 324 ms). This −0.98 μV difference was statistically significant, $t(43) = 5.96$, $p = < 0.001$, $d = 0.90$. These results indicate that the P2a, SDP, and N2pc were present on Go trials. An SDP was also observed on No-Go trials (0.71 μV), $t(43) = 5.88$, $p = < 0.001$, $d = 0.89$, but it was markedly reduced relative to that observed on Go trials, $t(43) = 11.00$, $p < 0.001$, $d = 1.66$ (**Fig 2B and 2C**). No N2pc was evident on No-Go trials, $t(43) = 0.84$, $p = 0.405$, $BF_{01} = 4.39$. Instead, the singleton was found to elicit ERP components associated with perceptual suppression and response inhibition: the $P_D$ (412 to 462 ms; 0.56 μV), $t(43) = 4.01$, $p < 0.001$, $d = 0.60$, and the no-go P3 (260 to 310 ms; 1.98 μV), $t(43) = 3.74$, $p < 0.001$, $d = 0.56$, respectively. As expected [32], the $P_D$ was evident only after the conventional N2pc time interval. The late onset of this $P_D$ indicates that observers initially ignore the orientation singleton without suppressing it proactively but that suppression is ultimately involved in preventing the distractor from accessing WM.

## ERP activities associated with target enhancement predict visual WM capacity

Our primary objective was to determine whether greater activation of attention processes associated with target enhancement—as reflected by increased amplitudes of the SDP and N2pc components—would predict higher WM capacity. To this end, we plotted participants' WM capacities as a function of their attention-control activities, separately for each ERP component and computed the Pearson correlation coefficient for each bivariate pairing. The coefficient was multiplied by −1 for the N2pc so that, in each case, a positive correlation would indicate that a larger ERP amplitude (positive or negative) was associated with higher WM capacity. Critically, individual participants' WM capacities (mean $K$: 3.0; range: 1.3 to 4.8) correlated positively with their SDP amplitudes, $r(43) = 0.37$, $p = 0.015$, and with their N2pc amplitudes, $r(43) = 0.35$, $p = 0.020$ (**Fig 3B and 3C**). To help visualise these relationships, we rank-ordered participants based on their WM capacities and then plotted the SDP and N2pc for separate subgroups of individuals ($n = 15$ each) with the highest and lowest capacities (**Fig 3D**). Unsurprisingly, the SDP and N2pc were visibly larger for the high-capacity group than for the low-capacity group. These results indicate that the target-enhancement processes driving the SDP and N2pc help to control the flow of visual information to WM systems.

No linear association was found between WM capacity and the amplitude of the P2a, $r(43) = 0.09$, $p = 0.554$, $BF_{01} = 4.49$ (**Fig 3A**). This indicates that high-capacity individuals are no more capable than their low-capacity counterparts at distinguishing between relevant-colour and irrelevant-colour arrays (but are more capable at engaging in subsequent search for a salient singleton, as indicated by the SDP and N2pc results). Interestingly, neither the amplitude of the $P_D$ nor that of the no-go P3 was found to correlate with WM capacity, $rs(43) \leq 0.19$, $ps \geq 0.228$, $BF_{01}s \geq 2.63$ (**S1 Fig**). These findings indicate that the inhibitory processes driving the $P_D$ and the no-go P3 were not critically involved in preventing distractor information from accessing visual WM in the task used here.

## Correlation disappears when search can be automated

Our second objective was to determine whether the linear relationships observed in Experiment 1 would continue to hold in the absence of the Go/No-Go element. At the outset, we hypothesised that attention-control processes associated with target enhancement would predict visual WM capacity only when the task required online control on each trial to prevent automation of target selection (see Introduction). To test this hypothesis, we instructed a

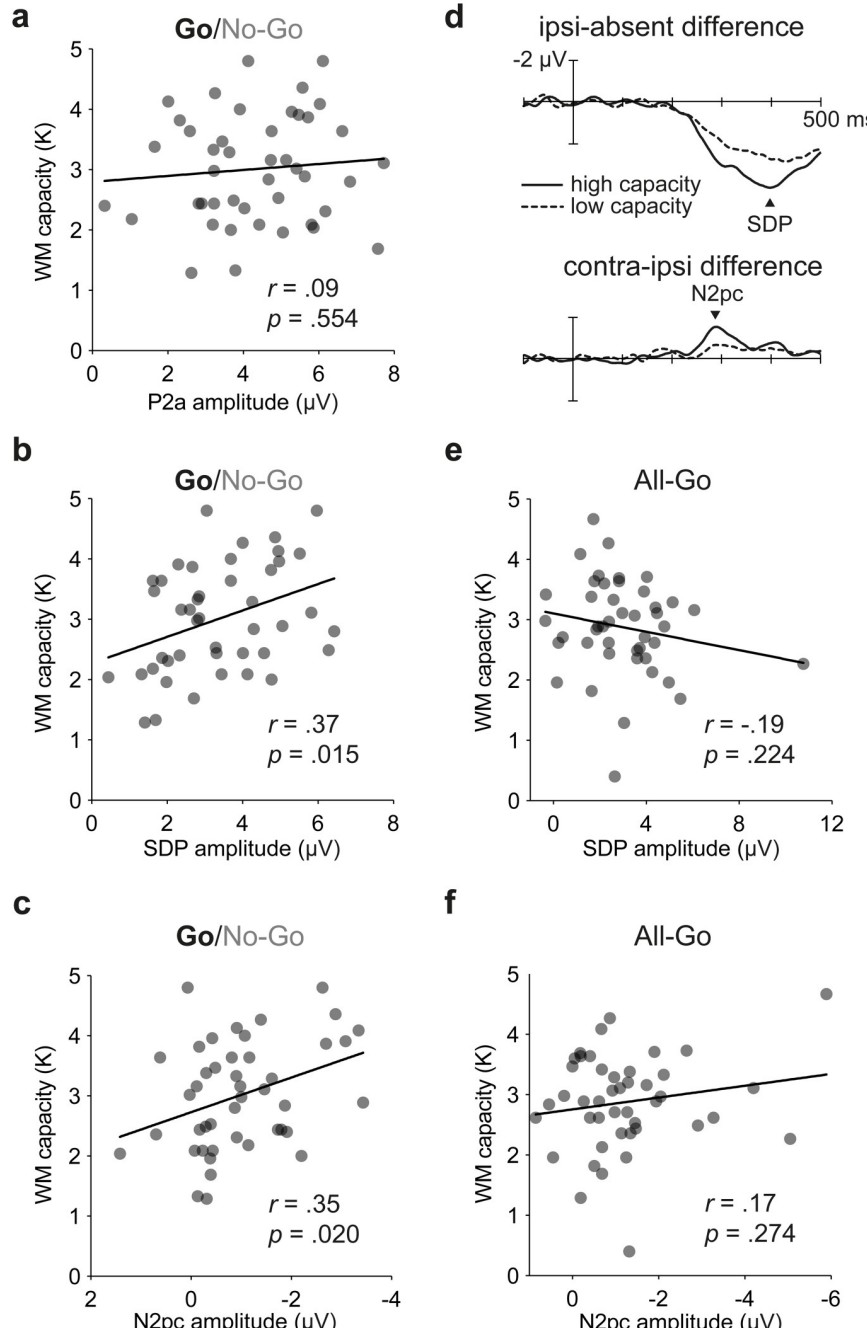

**Fig 3. Bivariate relations between individuals' WM capacities and amplitudes of isolated ERP indices of target-enhancement processes.** The underlying data supporting this figure can be found at https://osf.io/4wdzq. (**a**) Display-relevance activity over the frontal scalp (P2a) in Experiment 1 did not predict WM capacity. (**b**) Singleton-detection activity over the posterior scalp (SDP) on Go trials of Experiment 1 predicted WM capacity. (**c**) Attentional-selection activity over the posterior scalp (N2pc) on Go trials of Experiment 1 also predicted WM capacity. (**d**) On Go trials (Experiment 1), SDP and N2pc were larger for high-capacity group than for low-capacity group. Difference waves are from occipital electrodes PO7/PO8. (**e**) Singleton-detection activity over the posterior scalp (SDP) did not predict WM capacity in Experiment 2. (**f**) Attentional-selection activity over the posterior scalp (N2pc) did not predict WM capacity in Experiment 2. ERP, event-related potential; N2pc, posterior contralateral N2; P2a, anterior P2; SDP, singleton detection positivity; WM, working memory.

second group of 44 participants to search for singletons within both blue-item arrays and yellow-item arrays. Experiment 2 was similar to Experiment 1 apart from the instruction to indicate the presence or absence of the singleton on every trial (called All-Go trials). The occipital ERP waveforms resembled those from Experiment 1 (**Fig 2D**), except that the late positive deflections appearing approximately 200 ms after display onset were visibly smaller on both singleton-absent trials and singleton-present trials (no statistical tests were performed because this was not predicted in advance). Once again, the singleton-present waveforms were more positive than the singleton-absent waveform in the time range of the SDP, and the waveform contralateral to the singleton was more negative than its ipsilateral counterpart in the time range of the N2pc. Statistical tests indicated that singletons elicited both the SDP (2.98 μV) and the N2pc (−1.24 μV), $ts(43) \geq 6.01$, $ps < 0.001$, $ds \geq 0.91$ (**Fig 2D and 2E**). The N2pc occurred earlier on All-Go trials (179 ms) than on Go trials of Experiment 1 (261 ms), $t(86) = 5.76$, $p < 0.001$, $d = 1.23$, because participants did not make a Go/No-Go decision before searching for the singleton (see also [32]). Critically, the participants' WM capacities (mean $K$: 2.9; range: 0.4 to 4.7) did not correlate with the magnitudes of their SDP, $r(43) = −0.19$, $p = 0.224$, $BF_{01} = 2.60$, or their N2pc, $r(43) = 0.17$, $p = 0.274$, $BF_{01} = 2.98$, in Experiment 2 (**Fig 3E and 3F**). The split-half reliability estimates were high for the SDP and N2pc in Experiment 2 (Spearman–Brown coefficients of 0.92 and 0.79, respectively), which indicates that the absence of statistically significant correlations with WM capacity were not due to poor reliability of the ERP measures. Taken together, the findings from Experiments 1 and 2 indicate that low-capacity individuals have difficulty initiating pop-out search when online control is required on a trial-by-trial basis (Experiment 1) but not when the search processes can be automated (Experiment 2).

## A look at behavioural performance

Finally, although the Go/No-Go task was designed to reveal effects of WM capacity on isolated ERP measures of attentional control, we also assessed the behavioural performance measures from the 2 experiments. In Experiment 1, participants withheld responses on all but 0.12% of the No-Go trials on average, with 24 participants managing to fully comply with instructions to respond only on Go trials. Together with the ERP results reported above, this finding indicates that participants managed to terminate the processing of irrelevant-colour displays before the stages associated with searching and responding. Given the lack of variability in No-Go responses, we did not test for a correlation between the proportions of No-Go errors and WM capacity. The remaining analyses focused on singleton-present trials on which participants made correct responses, since these were the same trials used to study the neural mechanisms of selective target enhancement. The grand-averaged response times (RTs) were longer for Go trials of Experiment 1 (622 ms) than for All-Go trials of Experiment 2 (569 ms), $t(86) = 4.80$, $p < 0.001$, $d = 1.02$, because of the additional time required to evaluate the colour of the display (**Fig 4A**). However, the individual participants' mean RTs did not correlate with WM capacity in either experiment, $rs(43) \leq −0.13$, $ps \geq 0.394$, $BF_{01}s \geq 3.74$ (**Fig 4B and 4C**). The null result from Experiment 2 is consistent with the ERP results from that experiment and with the notion that automatic visual-search processes are insensitive to variations in WM capacity [23,26,38]. The null result from Experiment 1 is somewhat more surprising but is in line with null results from a previous study using a more conventional Go/No-Go task (respond to "X" but not to other letters) [39].

## Discussion

WM capabilities are known to affect performance in tasks that require maintenance and updating of relevant information, particularly in the presence of irrelevant information [6–10,14]. Several

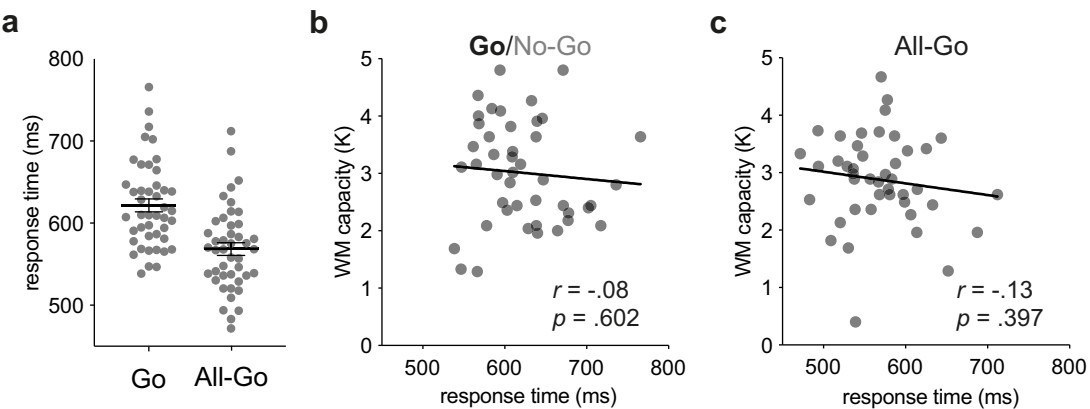

**Fig 4. RT results from Experiments 1 and 2.** The underlying data supporting this figure can be found at https://osf.io/4wdzq. (**a**) Mean RTs for correct singleton-present trials of Experiments 1 (Go trials) and 2 (All-Go trials). Each dot represents a participant's mean RT, and each horizontal line with SEM bars shows the grand-averaged RT. (**b**) Bivariate plot with WM capacity in Experiment 1. (**c**) Bivariate plot with WM capacity in Experiment 2. RT, response time; WM, working memory.

theoretical perspectives have emphasised the importance of executive-attention mechanisms for controlling what information gains access to visual WM and for maintaining focus on relevant information in tasks that require WM [9,10]. Many of these perspectives focus on inhibitory attention-control processes that filter out irrelevant sources of information that have the potential to interfere with an observer's task at hand [12–14,40–42]. The earliest and most influential of these perspectives—the inhibitory control theory of WM [40–42]—emphasises inhibition not because attention control is presumed to operate exclusively to suppress irrelevant information but because the control processes acting to enhance relevant information are assumed to be too automatic to be a factor in differentiating low- and high-capacity individuals [13].

Converging lines of evidence have confirmed the presumed link between WM capacity and inhibitory attention-control mechanisms, but to date no such link has been established for attention mechanisms that selectively enhance target processing. Behaviourally, low- and high-capacity individuals perform similarly across a variety of visual search tasks that are hypothesised to require focal attention to find the target [43]. Electrophysiologically, at least 3 studies reported to find no link between individual differences in WM capacity and the amplitude of the target-elicited N2pc [17,18,21]. This pattern of empirical results is consistent with the inhibitory control theory of WM capacity [13,14,40–42] as well as the more recent filtering-efficiency hypothesis, which attributes individual differences in WM capacity to differences in distractor-filtering capabilities [12,16]. Here, however, it was hypothesised that such a link would emerge in a Go/No-Go search task that prevented automation of target-selection processes. Results of the 2 present experiments were consistent with this hypothesis. The magnitudes of 2 target-elicited ERP components, the SDP and N2pc, were found to predict individual differences in visual WM capacity when to-be-searched displays and to-be-ignored displays were randomly intermixed across trials (Experiment 1). No such correlation was evident when participants were instructed to search for a target singleton on each and every trial (Experiment 2). Neither the SDP nor the N2pc could be attributed to distractor-filtering processes because the task was designed to prevent such filtering [19]. On the basis of these findings, we conclude that "excitatory" attentional mechanisms—ones that boost target processing rather than suppress distractor processing—help to control access to WM but fail to do so when target selection can be automated.

The results of the present study, and the conclusion stated above, have implications for existing theories of WM capacity that attribute capacity differences to differences in some

specific attention-control process(es). Nearly all of these theories are based on the observation that high-capacity individuals perform better than low-capacity individuals when task-relevant information is presented along with irrelevant information that might capture attention or otherwise interfere with performance. Most of these theories differ in whether WM capacity differences are attributable to inhibitory attention-control processes that prevent irrelevant information from consuming WM resources [12–14] or to executive-control processes that actively maintain or enhance relevant information in the face of potential distraction [22,24]. Consistent with the inhibitory control theory, past findings suggest that attention processes involved in target selection are too automatic to contribute to differences between high- and low-capacity individuals [17,18,21,27,43]. The present study demonstrated that target-centered attention processes contribute to capacity differences, but only when the task prevents automation of such processes. Inhibitory-control theories of WM need to be updated to permit a contribution from excitatory attention-control processes under such conditions.

The present findings are largely consistent with the executive attention theory of WM capacity, however. According to this theory, capacity is determined not by inhibitory processes but by attention processes that can be used flexibly to maintain task-relevant information or to suppress irrelevant information [22–24]. As noted by Unsworth and colleagues [23], this latter perspective predicts that individual differences in WM capacity will be evident in tasks that require controlled attention even when there is no need to inhibit. Consistent with this prediction, they found that low- and high-capacity individuals differ in their ability to make pro-saccades (i.e., saccadic eye movements toward an abruptly appearing visual stimulus), but only when pro-saccades trials were randomly intermixed with anti-saccade trials (on which saccades are made away from the stimulus). The mixed-trials design was presumed to increase the need for control of an otherwise automatic overt-orienting behaviour in the same way that the Go/No-Go search design was presumed to increase the need for control of an otherwise automatic covert-orienting process. Thus, the current electrophysiological findings buttress the conclusion that was based on performance in saccade tasks: Low- and high-capacity individuals differ in the control of target selection processes even when there is little or no requirement to inhibit processing of distractors.

Experiment 1 of the present study utilised a task that required inhibitory control on No-Go trials, and so we must consider whether the observed link between target selection and WM capacity was dependent not on the increased need for attention control but on the need to inhibit. Although this alternative explanation cannot be ruled out entirely at present, it would seem unlikely for at least 3 reasons. First, low- and high-capacity individuals typically perform similarly in Go/No-Go tasks unless the rules for responding are sufficiently complicated (e.g., respond to M or W, but only if the last target was different) [32]. The task used in Experiment 1 had no prepotent response and simple response alternatives, and thus low-capacity individuals would not be expected to have an inhibitory control deficit in the present study. Second, neither the $P_D$ nor the no-go P3 was found to correlate with visual WM capacity in the present study, thereby confirming that low-capacity individuals exhibited no inhibitory-control deficit in the present study. Third, even if an inhibitory-control deficit went undetected in Experiment 1, the observed relationship between the amplitude of the target-elicited N2pc and WM capacity is opposite to what might be predicted from an inhibitory-control perspective. Specifically, if low-capacity individuals were less able to inhibit on No-Go trials, target selection on inter-mixed Go trials might be expected to be facilitated due to a reduction of lingering inhibition from previous trials. By this account, the target-elicited N2pc would be larger for low-capacity individuals than for high-capacity individuals due to the reduction of lingering inhibition across trials. In light of these considerations, we believe that increased need for control, not inhibition, was responsible for the observed relationship between target N2pc and WM

capacity. This conclusion could be tested in the future by replacing the Go/No-Go task with other dual-task designs that would prevent the automation of target-selection processes.

## Summary and conclusion

High-capacity individuals are more capable of filtering out irrelevant information than their low-capacity counterparts across a wide range of tasks. Here, we show that high-capacity individuals are also more capable of selectively enhancing task-relevant targets when to-be-searched displays are randomly intermixed with to-be-ignored displays. The findings are consistent with theories of WM capacity that emphasise controlled attention for the establishment of links between WM capacity and the lower-level selection processes that regulate the flow of information to neural systems that subserve WM. We conclude that links between WM capacity and either distractor suppression or target enhancement will arise only when the low-level selection process contributes substantially to the task at hand and cannot be automated. In the present study, distractor suppression was not critical for task performance, and thus suppression was not predictive of capacity. In competitive search paradigms that pit the target against a more salient distractor [17], target-selection processes are assumed to be automated (leading to no link between target N2pc and capacity), whereas distractor-suppression processes are assumed to be more controlled (leading to a link between $P_D$ and capacity in that paradigm). These assumptions are consistent with findings from a recent dual task study, wherein the $P_D$ was abolished during the attentional blink (while attention was still engaged on a previous target in a rapid stream of stimuli), whereas the magnitude of the target-elicited N2pc was unchanged [44].

## Materials and methods

### Participants

The Research Ethics Board at Simon Fraser University approved the research protocol used in this study. Ninety-four young adults were recruited to participate in the experiments reported in this paper. After giving informed consent, 45 volunteers participated in Experiment 1 and 49 volunteers participated in Experiment 2. Participants received either course credit as part of a departmental research participation system or $20. All participants reported normal or corrected-to-normal visual acuity and were tested for normal colour vision using Ishihara colour plates prior to participation. Participant data were excluded from further analyses if more than 30% of their trials were contaminated by ocular artifacts (rejection criterion set in advance). Data from 6 participants were excluded in total (1 from Experiment 1 and 5 from Experiment 2). Of the remaining participants, 44 participated in Experiment 1 (mean age: 20.5 years), 27 of which were female and 41 of which were right-handed. Experiment 2 also had 44 participants (mean age: 19.9 years), 28 of which were female and 38 of which were right-handed. These sample sizes were selected a priori to give us sufficient power (0.80) to detect a moderately large linear correlation ($r$ = 0.40; calculated using G*Power Version 3.1.9.6). This effect size was a conservative estimate informed by 2 studies that found correlation between $P_D$ amplitude and visual WM capacity in the range of $r$ = 0.43 to 0.59 [17,18]. Our assumption is that a similar effect magnitude would be observed for a correlation between N2pc amplitude and WM capacity.

### Apparatus

Both experiments were conducted in a sound-attenuated and electrically shielded chamber dimly illuminated by DC-powered LED lighting. A height-adjustable LCD monitor presented

stimuli at 120 Hz. Participants sat in a chair and viewed the monitor at a distance of approximately 57 cm and made their responses using a gamepad. A Windows-based computer controlled stimulus presentation and registered participants' button presses using Presentation software (Neurobehavioral Systems, Berkeley, California). A custom software (Acquire) recorded electroencephalogram (EEG) from a second, Windows-based computer, which housed a 64-channel A-to-D board (PCI-6071e, National instruments, Austin, Texas) that connected to an EEG amplifier system with an input impedance of 1 GΩ (SA Instruments, San Diego, California). The stimulus-control and EEG-acquisition computers were situated outside of the testing chamber.

## Stimuli and procedure

**Experiment 1.** Each stimulus display consisted of a small, white fixation cross ($0.3° \times 0.3°$; $0.3$ cd/m$^2$) positioned at the middle of the display and 16 cyan ($0.3° \times 1.0°$; x = 0.20, y = 0.35, 17.5 cd/m$^2$) or 16 yellow lines ($0.3° \times 1.0°$; x = 0.37, y = 0.57, 28.0 cd/m$^2$) that appeared within a $11.1° \times 8.3°$ region around fixation. The coordinates of the lines were determined randomly, with the restrictions that all displays contain 8 lines on either side of fixation without crossing the horizontal or vertical meridians and that no lines connect or overlap. Singleton-absent displays contained 16 horizontal or 16 vertical lines. Singleton-present displays were identical to singleton-absent displays except one of the 16 lines was replaced with a line of an orientation orthogonal to that of the surrounding lines. The resulting 8 types of displays (colour × singleton presence × orientation) were randomly intermixed and presented with equal probability. Each display was presented for 750 ms, and the time between stimulus onset varied randomly between 1,350 ms and 1,650 ms. The colour of the lines indicated whether a given trial was Go or No-Go. For half of the participants, the cyan displays were used for Go trials and the yellow displays were used for No-Go trials. The colour assignment was reversed for the remaining participants. On Go trials, participants were asked to indicate the presence or absence of the singleton by pressing either the left or right shoulder button on a gamepad using their index fingers. The stimulus-response mapping was counterbalanced across participants. On No-Go trials, participants simply waited for the trial to end without providing a response. Each participant completed 40 blocks of 40 trials, yielding a total of 1,600 trials.

**Experiment 2.** The stimuli and procedure in Experiment 2 were identical to those in Experiment 1 except participants responded to both cyan and yellow stimulus displays and the entire experiment comprised 20 blocks of 40 trials for a total of 800 trials.

## Working memory capacity

Before each main experiment, participants completed a change-detection task that assessed their WM capacity. All stimuli and procedure for this task were identical to those used by ref. [7]. Briefly, participants viewed a sequence of displays on each trial, starting with a memory display lasting 150 ms. In the memory display, coloured squares of varying set sizes (2, 4, 6, 8) appeared in one of 36 possible locations (9 in each quadrant), the coordinates of which formed a regular grid. This display was followed by a 900-ms retention interval, during which only a fixation cross was presented at the centre of the display. Following this interval, a test display presented a coloured square at one of the locations previously occupied in the memory display. Participants pressed a button to indicate whether the square occupying that location changed in colour across the 2 displays. Each participants completed a total of 120 trials. Visual WM capacity ($K$) was computed separately for the set sizes of 4, 6, and 8 using a standard equation [2,45]. The resulting $K$ scores were then averaged to compute an estimate of individuals' WM capacity.

## Electrophysiological recording

EEG signals were recorded from 25 sintered Ag/AgCl electrodes housed in an elastic cap. The electrodes were positioned at standard 10–10 sites (FP1, FPz, FP2, F7, F3, Fz, F4, F8, T7, C3, Cz, C4, T8, P7, P3, Pz, P4, P8, PO7, POz, PO8, O1, Oz, O2, M1) and were referenced to an electrode positioned on the right mastoid during recording. The horizontal electrooculogram (HEOG) was recorded using 2 additional electrodes placed 1 cm from the external canthus of each eye and referenced to each other. The ground electrode was positioned over the midline frontal scalp at site AFz. The HEOG was used to detect eye movements away from the fixation cross. Eye blinks were monitored using the FP1 electrode. All electrode impedances were kept below 15 kΩ. EEG and EOG signals were amplified with a gain of 20,000, filtered using a band-pass filter of 0.01 to 100 Hz (two-pole Butterworth), and digitised at 500 Hz. The EEG signals were processed using the Event-Related Potential Software System (U. California San Diego, California). A semiautomated procedure was performed to remove epochs of EEG that were contaminated by horizontal eye movements, blinks, or amplifier blocking using our standard lab procedures [32]. Artifact-free data were then low-pass filtered (half-power cutoff) at 30 Hz to create averaged ERP waveforms. Each EEG channel was digitally rereferenced to the average of the left and right mastoid channels. The grand-averaged event-related EOG deflections were required to be below 2 μV for further inclusion of the data in the analysis. Positive voltages were plotted downward by convention.

## Analysis

**Experiment 1.** Approximately 3.7% of trials were excluded from all analyses due to incorrect responses (misses, false alarms, or no button presses on Go trials and button presses on No-Go trials). Of the correct-response trials, 0.2% were excluded because responses were too fast (response time; RT < 100 ms) or too slow (RT > 1,350 ms). Of the remaining trials, 10.2% were excluded because an artifact was detected in the electrophysiological recordings. Artifact-free ERPs were computed separately for singleton-present and singleton-absent displays and were further subdivided for Go and No-Go trials. For singleton-present displays, ERPs recorded contralateral and ipsilateral to the singleton were constructed using conventional methods (by collapsing across left- and right-field stimuli and left- and right-hemisphere electrodes).

All electrode sites used for the ERP measurements reported herein were chosen a priori based on where they were previously observed to be largest and to maintain consistency with prior studies [31,32]. The mean amplitude of each component was measured in 3 steps. First, the local peak amplitude of each component was determined using a relatively wide window that was chosen a priori based on previous literature (P2a: 150 to 300 ms; N2pc: 170 to 300 ms; $P_D$: 200 to 500 ms; no-go P3: 200 to 500 ms). Second, the time point at which each component first reached 75% of its peak amplitude was determined. Third, the mean amplitude of each component was measured in a 50-ms window (100-ms window for the longer-lasting SDP) that began at the latency determined in the previous step.

Each mean-amplitude measurement was taken from an appropriate difference waveform. The P2a and no-go P3 were isolated by subtracting ERPs elicited on No-Go trials from ERPs elicited on Go trials. The P2a was measured at FPz using a 186- to 236-ms window [32,33], and the no-go P3 was measured at Cz using a 260- to 310-ms window [32,46]. Here, the no-go P3 appeared as a negative deflection rather than as a positive deflection because the direction of the subtraction was opposite to that typically used to investigate no-go activity. The SDP was isolated by subtracting singleton-absent ERPs from singleton-present ERPs at electrodes PO7 and PO8 using a 318- to 418-ms window [31,32,47]. This measurement was performed

only on the ipsilateral difference waves because the magnitude and timing of the contralateral SDP would be obscured by the N2pc or $P_D$. The N2pc and $P_D$ were isolated by subtracting ipsilateral ERPs from corresponding contralateral ERPs at PO7 and PO8. The N2pc was measured on Go trials for singletons in the lower field using a 274- to 324-ms window [31,32,47], and the $P_D$ was measured on No-Go trials for singletons in the upper field using a 412- to 462-ms window [20,32].

All statistical tests reported herein were performed with 2 tails using JASP (version 0.16.1). Furthermore, because of the inherent difficulty in asserting null hypotheses using conventional tests, we computed the Bayes factor (BF) following all nonsignificant tests. A default scale r (Cauchy scale) value of 0.707 was used to compute BFs. We reported $BF_{01}$ values to denote the likelihood of observing the data given the null hypothesis is true relative to observing the data given the alternative hypothesis is true.

Presence of each ERP component was assessed using one-sample $t$ tests against 0 μV, separately for Go and No-Go trials. To assess for linear relationships between participants' WM capacity and their excitatory control processes, we computed Pearson correlation coefficients between $K$ and mean amplitudes of the P2a, SDP, and N2pc elicited on Go trials. To assess for linear relationships between participants' WM capacities and inhibitory control processes, we computed Pearson correlation coefficients between $K$ and mean amplitudes of the $P_D$ and no-go P3 elicited on No-Go trials. The signs of the obtained Pearson correlation coefficients for negative-voltage components (i.e., the N2pc and no-go P3) were reversed (i.e., multiplied by −1) so that a positive coefficient would indicate that larger ERP negativities were associated with higher WM capacity.

Behavioural performance in the present experiment was measured in 2 ways. First, response error rates of individual participants on No-Go trials were computed. Second, mean RTs of singleton-present displays on Go trials were computed for each participant. Singleton-present displays were specifically chosen to match the trials we used to study the neural mechanisms of excitatory attention. To assess for a linear relationship between WM capacity and behavioural performance, Pearson correlation coefficient was computed between $K$ and the mean RTs. A correlation between WM capacity and No-Go response errors was not evaluated due to more than 54% of the participants making no such errors.

**Experiment 2.** Approximately 7.3% of trials were excluded from all analyses due to incorrect responses (misses, false alarms, or no button presses). Of the correct-response trials, 0.7% were excluded because responses were too fast (RT < 100 ms) or too slow (RT > 1,350 ms). Of the remaining trials, 9.9% were excluded because an artifact was detected in the electrophysiological recordings. Artifact-free ERPs were computed separately for singleton-present and singleton-absent displays. The method for isolating the SDP and N2pc were identical to that in Experiment 1. No other ERP components were isolated or measured. Mean amplitudes of the SDP and N2pc were measured in a 268- to 368-ms window and a 212- to 262-ms, respectively. In addition, latencies of the N2pc in this experiment and that elicited on Go trials of Experiment 1 were computed as the time point at which they first reached 50% of their peak amplitude, using the conventional jackknife procedure [48,49]. As in Experiment 1, all statistical tests were performed with 2 tails, and $BF_{01}$ values were computed following all nonsignificant tests. Presence of the SDP and N2pc was assessed using one-sample $t$ tests against 0 μV. Latency of the N2pc elicited in the present experiment was then compared with latency of the N2pc elicited on Go trials of Experiment 1 using independent-samples $t$ tests. Linear relationships between participants' WM capacities and magnitudes of their SDP and N2pc were assessed by computing Pearson correlation coefficients between $K$ and mean amplitudes of the SDP and N2pc. The Pearson correlation coefficient for the N2pc was multiplied by −1 so that a positive correlation would indicate that an increase in N2pc was associated with larger WM capacity.

As in Experiment 1, behavioural performance was measured in terms of mean RTs of singleton-present displays. To assess for linear relationships between WM capacity and behavioural performance, Pearson correlation coefficient was computed between $K$ and mean RTs of singleton-present displays.

## Supporting information

**S1 Fig. Bivariate relations between individuals' WM capacities and amplitudes of isolated ERP indices of inhibition in Experiment 1.** The underlying data supporting this figure can be found at https://osf.io/4wdzq. (**A**) Distractor-suppression activity over the posterior scalp ($P_D$) on No-Go trials of Experiment 1 did not predict WM capacity. (**B**) Response-suppression activity over the central scalp (no-go P3) on No-Go trials of Experiment 1 did not predict WM capacity.
(PDF)

## Acknowledgments

We thank Juliet Fowler, Alex Nash, and Leanne Vibar for assistance in data collection.

## Author Contributions

**Conceptualization:** Daniel Tay, John J. McDonald.

**Formal analysis:** Daniel Tay, John J. McDonald.

**Funding acquisition:** John J. McDonald.

**Investigation:** Daniel Tay.

**Methodology:** Daniel Tay, John J. McDonald.

**Project administration:** Daniel Tay.

**Software:** Daniel Tay, John J. McDonald.

**Supervision:** John J. McDonald.

**Visualization:** Daniel Tay, John J. McDonald.

**Writing – original draft:** Daniel Tay, John J. McDonald.

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
