## [Editor Report · Decision Letter 0]

29 Jul 2022

Dear Dr Tay, 

Thank you for submitting your manuscript entitled "Excitatory attention control activity predicts individual differences in visual working memory" for consideration as a Short Reports by PLOS Biology.

Your manuscript has now been evaluated by the PLOS Biology editorial staff, as well as by an academic editor with relevant expertise, and I am writing to let you know that we would like to send your submission out for external peer review.

Once your full submission is complete, your paper will undergo a series of checks in preparation for peer review. After your manuscript has passed the checks it will be sent out for review. To provide the metadata for your submission, please Login to Editorial Manager (https://www.editorialmanager.com/pbiology) within two working days, i.e. by Jul 31 2022 11:59PM.

Kind regards,

Kris

Kris Dickson, Ph.D. (she/her)

Neurosciences Senior Editor/Section Manager

PLOS Biology

kdickson@plos.org

---

## [Decision Letter · Decision Letter 1]

20 Sep 2022

Dear Dr Tay,

Thank you for your patience while your manuscript "Excitatory attention control activity predicts individual differences in visual working memory" was peer-reviewed at PLOS Biology. It has now been evaluated by the PLOS Biology editors, an Academic Editor with relevant expertise, and by three independent reviewers. 

In light of the reviews and additional feedback from our Academic Editor, all of which you will find at the end of this email, we are inviting you to revise the work to thoroughly address the reviewers' reports. However, as you will see below, the reviewers raise a number of concerns related to the overall impact of this work as the study is currently laid out. Because of this, we would be looking to see that the reviewers and our Academic Editor were convinced that the revisions made a clear and convincing case for the impact of this work and its suitability for PLOS Biology. 

Given the extent of revision needed, we cannot make a decision about publication until we have seen the revised manuscript and your response to the reviewers' comments. Your revised manuscript is likely to be sent for further evaluation by all or a subset of the reviewers.

**IMPORTANT - SUBMITTING YOUR REVISION**

*Re-submission Checklist*

*Published Peer Review*

*PLOS Data Policy*

*Blot and Gel Data Policy*

Sincerely,

Kris

Kris Dickson, Ph.D. (she/her)

Neurosciences Senior Editor/Section Manager

PLOS Biology

kdickson@plos.org

REVIEWS:

Do you want your identity to be public for this peer review?

Reviewer #1: No

Reviewer #2: Yes: Sirawaj Itthipuripat

Reviewer #3: No

Reviewer #1: Synopsis: Tay & McDonald report two experiments designed to test whether excitatory control processes correlate with working memory (WM) capacity, which would challenge prevailing views of WM gating based on inhibitory control. The authors track a multitude of ERP components linked with excitatory and inhibitory attentional control in previous studies while participants perform a go/no-go variant of a pop-out search task (Exp 1) or a classic all-go pop-out search task (Exp 2). Since the former task requires a greater degree of control than the latter (i.e., go/no-go plus target detection vs. just target detection), a link between WM capacity and ERP correlates of excitatory control processes should be most evident there. This is borne out in the experiments; two excitatory ERP components predict WM capacity in Experiment 1, but not Experiment 2. Thus, excitatory control processes also play a role in gating access to WM. 

Evaluation: This is a technically sound study that makes an important point about mechanisms that control access to WM. The conclusions hinge on whether the ERP components are linked to excitatory and inhibitory control processes in the manner the authors assume. I'm not fully up-to-date with the ERP literature, but each claim made by the authors is supported by at least one prior published study and I'm prepared to take the authors' interpretations on faith (other referees can perhaps speak to any shortcomings here). My only concern is that the authors might be underselling their conclusions: prevailing models of WM input gating focus exclusively (to my knowledge) on the operation of inhibitory mechanisms that "filter out" task-irrelevant information; the current data argue against this view. This is only very briefly discussed in the last paragraph of the paper. If space allows, I encourage the authors to expound on their suggestion that "Current attentional-control perspectives of WM need to be updated to account for these findings". For example, which perspectives are the authors referring to, and how - in the authors' view given the data - should they be modified?

Reviewer #2 (Sirawaj Itthipuripat): In the present paper, Tay and McDonald examined the relationship between the excitatory attentional control processes and individual differences in working memory (WM) capacity. They argued that although both excitatory and inhibitory processes have been shown to underly attentional control, only the inhibitory processes have been associated with individual differences in WM capacity. In their version of the go-nogo visual search task, they found that the amplitudes of the ERPs that track the excitatory processes in the go trials, i.e., the SDP and the N2pc components (but not the P2a component), were positively correlated with intersubjective variability in WM capacity. In contrast, the inhibitory ERPs including the Pd and the P3 components in the nogo trials did not predict WM capacity across subjects. In a control study, where subjects searched for a pop-out target in every trial, they found no correlation between the SDP and N2pc amplitudes and WM capacity. Taken together, they proposed that the excitatory attentional processes control access for visual WM. Overall, I think the study addressed a novel and important question that has a potential to shape theories that explain the interaction between attention and WM mechanisms. The experiments were well designed and executed with appropriate ERP and statistical analyses. The results provide new mechanistic insights into how different types of neural computations underling attentional selection may interact with WM. I only have some minor concerns. 

1) Even though the question is current and interesting, I think the paper needs a better motivation—not just that past studies only showed evidence for inhibitory processes and now we simply want to see if excitatory processes are involved. There should be a clearer hypothesis-driven motivation e.g., why excitatory attentional control processes are important for WM, how their involvement might be functionally distinct from the inhibitory processes, and at what context one mechanism might me more dominant than the other. 

2) Related to (1), why in some contexts (like in past studies) inhibitory mechanisms dominate, and in some contexts (like in the present study) excitatory processes dominate. This should be discussed further. What special about the nogo component of the task that made the results diverted from past studies that observed the more dominant role in the inhibitory processes?

3) Why the P2a component, which is also excitatory in the authors' view, did not predict WM capacity like the N2pc and SDP did. This should be discussed in detail. 

4) I think the reader would benefit from seeing the topographic maps of different ERP components.

5) I would love to see additional figures showing correlation results between Pd/P3 and WM capacity to ensure that null results were not driven by some outliers. Right now, there were only correlation figures for the P2a, N2pc and SDP data.

6) When introducing excitatory and inhibitory mechanisms for the first time (in the abstract), I think the authors should be more specific by relating those mechanisms to the attentional processes. Without no background, the reader might get confused that the authors meant excitatory and inhibitory mechanisms that directly underly WM rather than attention itself.

7) Can the authors discuss limitations of using ERPs to track excitatory and inhibitory processes? Since ERPs are the population-level neural responses, how could the authors be sure that certain ERP components are truly excitatory or inhibitory? It would be good to cite non-human primate work that provides the links between the single unit activity and population-level activity as well as the associated excitatory and inhibitory processes.

Signed Sirawaj Itthipuripat

Reviewer #3: "Excitatory attention control activity predicts individual differences in visual working memory," Tay and McDonald. In two experiments, subjects first performed colored-blocks-array 'change detection,' to estimate working memory (WM) capacity, then a pop-out search task while the EEG was recorded concurrently. In Exp 1, trials varied unpredictably between conventional trials ("Go") and "No-Go" trials during which they were to withhold their response. Exp 2 featured only Go trials. The authors focus on three 'excitatory' components (P2a; SDP; and the N2pc) and two 'inhibitory' components (PD; and no-go P3). The motivation for this study is that although there is a great deal of evidence linking inhibitory processes to individual differences in WM capacity, the same is not true for "excitatory" processes. The authors "surmised [that] a link between excitatory attention control and WM capacity would be revealed in a task that requires more control than the conventional pop-out search paradigm," and the Go/No-Go procedure from Exp 1 was intended to engage this control.

The results are generally consistent with the authors' surmise, although I'm not sure that they are as impactful as they are made out to be. Part of the problem I'm having may come from a disconnect between the rather "high" theoretical level that is engaged to motivate this work versus the very concrete, mechanistic dependent measures that are collected and interpreted. That is, the theoretical framing is at a relatively high level that treats the constructs of "inhibition" and "excitation" as latent variables, and invokes papers like Redick et al. (2011), Hasher et al. (1999), and Kane et al. (2001) that conceptualize "inhibition" as a trait*. When engaging at this level, it can make sense to make a statement like "the gateway to WM is predominantly inhibitory in nature." However, these experiments don't engage directly with this "higher" theoretical level. Rather they engage a mechanistic/implementational level of specific processes, the processes associated with the 5 ERP components summarized above. At this more mechanistic level, a statement like "the gateway to WM is predominantly inhibitory in nature" simply doesn't make sense, because it has to be the case that the encoding of information into WM requires some operation akin to selection or input gating. A much more theoretically coherent way to motivate this paper would be with a sentence from the final paragraph of the main text (the "Discussion" paragraph): "… prior studies reported that WM does not vary with amplitude of the target-elicited N2pc." It's perhaps less grandiose, but it's also a more accurate reflection of what's at stake in this paper. (Doing this would also require tempering the final sentence about these needing to update current attentional-control perspectives of WM.)

I do think that it is useful, as a demonstration of specificity, to show that the "inhibitory" ERPs (the PD** and the no-go P3) do not show the same dependence on block homogeneity as do the "excitatory" ones. It's conceptually awkward, however, because elsewhere it's stated that "the link between WM capacity and inhibitory attention control has been well established." (It is also puzzling, given this, why it's stated that "Our second objective was to determine whether the inhibitory-control activity … would also predict WM capacity.") Should one infer from these null results that these two components are not good indices of inhibitory attention control? Alternatively, I think one could argue that there's no reason to expect strong effects with this design, because subjects have no reason to actively suppress individual items (i.e., no reason prevent their selection) because no-go trials only require withholding a response.

The final thing I'll note is that the assertion that "excitatory" ERP effects haven't previously been associated with WM capacity feels somewhat disingenuous in that it overlooks the huge (and hugely influential) literature on the CDA. Indeed, I think that one way to make these results about more than 'just' the empirical question of whether one can or cannot obtain an association between the SDP and the N2pc and WM capacity would be to speculate about how the positive findings from this study fit into the literature on the CDA.

General note: The transition from Introductory Paragraph to the Main Text is very abrupt. This might be a constraint of the short-report format, but a few sentences of introduction delving into the design might improve overall readability. 

Methodological question: It doesn't make sense to me that one would include the capacity estimated at set-size 2 in the values that are averaged to estimate an individual's overall capacity. The theoretical upper limit at SS2 is 2, so its inclusion would necessarily underestimate the k of a subject whose true k is > 2.

* Note, however, that if one is reasoning from this higher level there are other perspectives that should also be taken into account, such as Unsworth, N., Fukuda, K., Awh, E., & Vogel, E.K. (2014). Working memory and fluid intelligence: Capacity, attention control, and secondary memory. Cognitive Psychology, 71, 1-26.

**why is it referred to as just "PD" in some places and as "late PD" in others?

Academic Editor: 

Regarding the overall impact: Considering that the relationship between excitatory measures of attention & wm capacity is only observed in the go/no go procedure, couldn't the continual need to inhibit attention between trials (disengage from some; engage for others) be interpreted that the connection to wm capacity is still dependent on some form of inhibition? The authors briefly mention this possibility, but dismiss it because there were few "no-go" errors. I don't this is a sufficient argument because their neural measures of attention are so much earlier than behavioral responding. I'd like to see them grapple with this issue a bit more explicitly.

Technically: A key part of the authors' argument relies on their ability to interpret a null correlation in Experiment 2. This can be problematic for a number of reasons, particularly if one or more of the measurements has low reliability. That is, the strength of a correlation between two variables cannot be higher than the reliability level of the either of the measures. The authors need to calculate and report the reliability of each of their measures to show that the lack of correlation in Exp 2 wasn't simply because they had poor reliability for their measurements.

---

## [Editor Report · Decision Letter 2]

7 Nov 2022

Dear Dr Tay,

Thank you for your patience while we considered your revised manuscript "Attentional enhancement of visual-search target predicts individual differences in visual working memory" for publication as a Short Reports at PLOS Biology. This revised version of your manuscript has been evaluated by the PLOS Biology editors and the Academic Editor.

Based on our Academic Editor's assessment that your revision nicely responded to the reviewer's concerns and now makes a clear case for the impact of the work within the existing literature, we are likely to accept this manuscript for publication. To move towards publication however, we need you to consider an editorial request and to address the following data and other policy-related requests.

***Title change:

Please consider modifying the title to emphasize the fact that you were able to uncover this attentional effect due to the specific task conditions used - i.e. the visual working memory task was constantly changing so couldn't become rote. Maybe something like:

"Attentional enhancement is predictive of individual differences in visual working memory under demanding conditions"

***Financial Disclosure:

Please include the appropriate grant numbers here.

***Data Availability:

Currently, your data is listed as "some restrictions will apply. Data are unsuitable for upload because they are stored in a custom format that can only be analyzed with custom lab software. However, data remain available upon request from the authors.” Access will need to be made open for us to proceed at PLOS Biology.

Note that we do *not* require all raw data. Rather, we ask that all individual quantitative observations that underlie the data summarized in the figures and results of your paper be made available in one of the following forms:

Fig 2 A-C; Fig3 A-F; Fig 4 A-C; SuppFig 1A-B

Please also ensure that figure legends in your manuscript include information on where the underlying data can be found (e.g. “The underlying data supporting Fig X, panel Y can be found in file Z.”)., and ensure your supplemental data file/s has a legend.

Please also ensure that your Data Statement in the submission system accurately describes where your data can be found.

We expect to receive your revised manuscript within two weeks. 

*Published Peer Review History*

*Press*

Sincerely,

Kris

Kris Dickson, Ph.D., (she/her)

Neurosciences Senior Editor/Section Manager,

kdickson@plos.org,

PLOS Biology

---

## [Editor Report · Decision Letter 3]

14 Nov 2022

Dear Dr Tay,

Thank you for the submission of your revised Short Reports "Attentional enhancement predicts individual differences in visual working memory under go/no-go search conditions" for publication in PLOS Biology. On behalf of my colleagues and the Academic Editor, Ed Vogel, I am pleased to say that we can in principle accept your manuscript for publication, provided you update your data deposition site (see below for more information) and that you address any remaining formatting and reporting issues. The formatting and reporting issues will be detailed in an email you should receive within 2-3 business days from our colleagues in the journal operations team; no action is required from you until then. Please note that we will not be able to formally accept your manuscript and schedule it for publication until you have completed all of these requested changes.

--------

***Data deposition:

Thank you for taking the time to deposit your data onto your university repository. Unfortunately, we cannot accept sole deposition of data to a non-static site, including institutional-based sites and GitHub. (https://journals.plos.org/plosbiology/s/data-availability). We therefore request deposition of your summary data to a static site, like Zenodo, FigShare or OSF. 

Once this change has been made, please also update your manuscript figure legends and the details section online to reflect this new deposition site.

--------

PRESS

We frequently collaborate with press offices. If your institution or institutions have a press office, please notify them about your upcoming paper at this point, to enable them to help maximize its impact. If the press office is planning to promote your findings, we would be grateful if they could coordinate with biologypress@plos.org. If you have previously opted in to the early version process, we ask that you notify us immediately of any press plans so that we may opt out on your behalf.

Sincerely, 

Kris

Kris Dickson, Ph.D., (she/her), (she/her)

Neurosciences Senior Editor/Section Manager

PLOS Biology

kdickson@plos.org